# Nanopore Technology Applied to Targeted Detection of Tomato Brown Rugose Fruit Virus Allows Sequencing of Related Viruses and the Diagnosis of Mixed Infections

**DOI:** 10.3390/plants12050999

**Published:** 2023-02-22

**Authors:** Raied Abou Kubaa, Serafina Serena Amoia, Giuseppe Altamura, Angelantonio Minafra, Michela Chiumenti, Fabrizio Cillo

**Affiliations:** 1Institute for Sustainable Plant Protection—National Research Council, 70126 Bari, Italy; 2Department of Soil, Plant and Food Sciences, University of Bari Aldo Moro, 70126 Bari, Italy; 3Research, Experimentation and Education Centre in Agriculture (CRSFA) “Basile Caramia”, Via Cisternino 281, 70010 Locorotondo, Italy

**Keywords:** high-throughput sequencing (HTS), nanopore sequencing, tomato brown rugose fruit virus (ToBRFV), tomato mottle mosaic virus (ToMMV), pepino mosaic virus (PepMV), *Tobamovirus*, plant virus detection, diagnostics, mixed infections

## Abstract

Tomato (*Solanum lycopersicum*) plants from a commercial glasshouse were identified with symptoms compatible with a tomato brown rugose fruit virus (ToBRFV) infection. Reverse transcription-PCR and quantitative PCR confirmed the presence of ToBRFV. Subsequently, the same RNA sample and a second from tomato plants infected with a similar tobamovirus, tomato mottle mosaic virus (ToMMV), were extracted and processed for high-throughput sequencing with the Oxford Nanopore Technology (ONT). For the targeted detection of ToBRFV, the two libraries were synthesized by using six ToBRFV sequence-specific primers in the reverse transcription step. This innovative target enrichment technology enabled deep coverage sequencing of ToBRFV, with 30% of the total reads mapping to the target virus genome and 57% mapping to the host genome. The same set of primers applied to the ToMMV library generated 5% of the total reads mapping to the latter virus, indicating that sequencing of similar, non-target viral sequences was also allowed. Further, the complete genome of pepino mosaic virus (PepMV) was also sequenced from the ToBRFV library, thus suggesting that, even using multiple sequence-specific primers, a low rate of off-target sequencing can usefully provide additional information on unexpected viral species coinfecting the same samples in an individual assay. These results demonstrate that targeted nanopore sequencing can specifically identify viral agents and has sufficient sensitivity towards non-target organisms to provide evidence of mixed virus infections.

## 1. Introduction

Tomato (*Solanum lycopersicum*) represents one of the most important crops worldwide, with a leading role for Italian production within the European and the international market [1]. Despite the species’ adaptability to different environmental conditions and rapid technological progresses, tomato production is under constant threat, since many pathogens including viruses are able to elude disease management strategies and cause substantial economic losses [2].

In commercial greenhouses, controlling the spread of viruses is particularly challenging. Inside greenhouses, viruses commonly derive from contaminated seed or poor sanitary conditions in plant nurseries [2,3]. Rapid outbreaks can occur due to the intensive manual practices (i.e., grafting, de-leafing, and bumble bee pollination), as are required in greenhouse tomato production systems [4]. Tomato brown rugose fruit virus (ToBRFV, genus *Tobamovirus*, family *Virgaviridae*) is an example of a mechanically transmitted virus that is rapidly changing its status from “emerging pathogen” to actual threat to tomato crops worldwide. Since its first description in the Middle East [5,6] ToBRFV was found successively in several countries, as recently reviewed [7]. ToBRFV has a single-stranded, positive-sense RNA genome containing four open reading frames (ORFs) that code for the small and large subunit of the viral replicase (RdRP, ORF 1 and ORF2 respectively), the movement protein (MP, ORF3), and the coat protein (CP, ORF4). Tobamoviruses affecting solanaceous crops include other closely related members such as tobacco mosaic virus (TMV), tomato mosaic virus (ToMV), and tomato mottle mosaic virus (ToMMV) [8,9,10]. Another widespread virus, affecting greenhouse tomato production in Europe and North America, is pepino mosaic virus (PepMV), in the genus *Potexvirus* and family *Alphaflexiviridae* [2,3,11].

Symptoms induced by ToBRFV in greenhouse-grown tomato plants are mostly aspecific and can vary within a wide range. Plants can be infected asymptomatically or show visible symptoms ranging from very severe to mild. PepMV and other tobamoviruses may cause similar leaf and fruit alterations that may be confused with ToBRFV symptoms [12,13]. Furthermore, as recently reported, ToBRFV and PepMV mixed infection can often exacerbate the disease phenotype, suggesting that the implementation of effective integrated management practices to prevent virus entry into the greenhouse environment is crucial for the control of new severe disease outbreaks [14].

In order to prevent the huge losses caused by ToBRFV, the European Commission, and the USA and China governmental authorities among other countries, issued emergency measures prohibiting the introduction and movement within their territories of *S. lycopersicum* and *Capsicum* spp. plant material, as they are considered the major host plants of ToBRFV. Such legislative acts enforce the undertaking of eradication measures in the case of confirmed presence of ToBRFV [12,15,16,17]. Since ToBRFV is a seedborne virus and its introduction in new areas is assumed to occur through virus-contaminated seeds [18], a fundamental disease management measure consists of employing virus-free seeds that are harvested from healthy tomato or pepper plants, or seeds that have been sanitized with effective chemicals [7]. Additionally, resistant genotypes have been sought for, and sources of promising genetic resistance to ToBRFV and other tobamoviruses are now available [19,20,21]. However, introgressing valuable resistance traits into commercial tomato genotypes by either conventional or biotechnological routes will be a laborious process that will take some years to be completed.

For all control measures, rapid and reliable diagnostic protocols are needed. Official ToBRFV detection methods for the European and Mediterranean Plant Protection Organization (EPPO) include both conventional and real-time RT-PCR protocols, while other molecular and serological tests have also been proposed [12]. The same EPPO source identifies high-throughput sequencing (HTS) technologies as useful for obtaining complete genome sequences, and also enabling the untargeted sequencing of total nucleic acids for the generic detection of diverse, previously unknown viruses affecting plant material under test [22]. Among available HTS platforms, Oxford Nanopore Technologies (ONT) recently introduced a third-generation sequencing technology that enables direct sequencing of long DNA or RNA molecules. ONT sequencing has several advantages, including single-molecule sequencing, longer sequencing read lengths (hundreds to thousands of nucleotides), high sequencing speed, and real-time monitoring of sequencing data [23,24]. Due to those benefits, ONT sequencing has been successfully introduced as a pathogen detection technology, and also in plant virology, as recently reviewed [23]. Previous nanopore sequencing experiments on ToBRFV-infected plant samples indicated that the adoption of the MinION platform could be advantageous for laboratory diagnosis and viral genome analysis [25,26]. In our laboratory, we used ONT sequencing to discover potential etiological viral agents affecting symptomatic *Jasminum officinale* plants. Either dsRNA or total RNA were used, and the presence of a member of the genus *Carlavirus* was identified [27].

The present work aims to evaluate the potential of the nanopore sequencing technology for ToBRFV detection and full-length genome characterization in infected tomato samples. We specifically targeted the virus genome through the reverse transcription step, by using different reverse primers annealing at fixed intervals along the ToBRFV sequence. We verified the specificity of this innovative approach by using the same primer set for sequencing from a second tomato sample infected with a different tobamovirus species. Finally, we performed the bioinformatics analysis for the identification of any tomato and viral sequences in our dataset, discovering a previously undetected mixed infection with PepMV.

## 2. Results

### 2.1. ToBRFV RT-PCR and RT-qPCR Detection

In May 2021, the emergence of a possible ToBRFV outbreak was discovered near Bari (Region Puglia, Southern Italy). From symptomatic plants, total RNA was extracted from leaf, peduncle, and fruit tissues. The presence of ToBRFV was detected in all tested samples by one-step RT-PCR, which revealed a DNA amplification product of the expected 560 bp size (not shown). The same RNA samples were then subjected to the duplex TaqMan real-time RT-PCR assay, for additional confirmation of the presence of ToBRFV according to the internationally acknowledged diagnostic protocol. Both FAM-labeled CaTa28 and Cy5-labeled CSP1325 probes produced an exponential amplification curve, denoted by Cq values between 20 and 30, thus confirming the presence of ToBRFV in the tomato samples (Figure 1).

### 2.2. Nanopore Sequencing of Virus-Infected Tomato Samples

The two ONT sequencing runs yielded 613,263 reads with an average length of mosaic 8 bp and average read quality of 8.56 phred score for the ToMMV library (bar03), and 703,328 reads with an average length of 615 bp for the ToBRFV library (bar04). After Epi2me filtering, a total of 530,100 reads with 437 nt average length and 624,419 reads with 625 nt average length were obtained from the ToMMV and ToBRFV libraries, respectively.

Quality-checked reads from both libraries were then de novo assembled using either Flye or Canu programs. Contig reconstruction produced a total of four and one contigs from the ToBRFV library with Flye and Canu, respectively. Two contigs were obtained with both assemblers using reads from the ToMMV library. Blastn annotation of the contigs together with other details are reported in Table 1.

Unexpectedly, PepMV was identified in the bar04 library, together with ToBRFV. Additionally, the tobamovirus species sequenced as target specificity control in the bar03 library was confirmed to be a ToMMV isolate (Table 1). Sequences whose accession numbers were obtained by blastn annotation were then used for a mapping to reference step. The tomato (including organelles) genome was also used as the reference, to evaluate the origin of other reads obtained during the sequencing. Mapping details are shown in Table 2. Remarkably, the use of six ToBRFV-specific reverse primers in the reverse transcription step (RP1-6, see Materials and Methods Section 4.3) allowed a significant enrichment of the target virus specific reads, that reached over 30% of the total reads. Conversely, in the same Bar04 library, PepMV sequences accounted for 2.5% of the total, while in the Bar03 library ToMMV reads did not reach 5% of the total (Table 2). The remaining nonviral reads mapped to the tomato genome and, in a range of 0.04–1.7%, to either the chloroplast or the mitochondion genome (Table 2).

### 2.3. ToBRFV Genome Reconstruction by Mapping of ONT Reads

The MinION sequencing run of the greenhouse-infected tomato sample produced 40,114 reads mapping on the ToBRFV genome (Table 2), with an average coverage depth of 6840.46× and a mean length of 806 nt. Individual reads were subjected to BLASTn/BLASTx searches against GenBank sequences in order to exclude reassortants. The complete genome sequence consisted of 6392 nt. Figure 2A illustrates the mapping of ToBRFV-specific reads on the reference genome. Poor coverage was found for reads mapping at the first 12 nucleotides (nt) at viral RNA 5′-end (9 reads), and at the last 17 nt (6381–6397, 2 reads) at the 3′-end of the reference ToBRFV genome. The highest read coverage was reached in the region spanning approximately nt 4300–5180 in the viral genome. This peak of sequences is compatible with a highly active reverse transcription process primed by RP5, that is complementary to the ToBRFV RNA approximately at position 5200 (Figure 2A). Individual reads showed some degree of nucleotide sequence variability which may represent the intrinsic low accuracy of the nanopore sequencing technique. Despite this variation, an almost complete consensus of the viral genome could be assembled, that shared 99% identity at nucleotide level to ToBRFV isolates from Jordan (Tom1-Jo, acc. no. KT383474) and Israel (IL, KX619418). When considering the alignment, reads presented nucleotide changes compared to the reference sequence, that were consistently detected in the vast majority of the reads for each position. Figure 2B illustrates an example where many point sequence variations in individual reads within a genome segment do not hinder the construction of a consensus sequence. In the same example (Figure 2B), a real substitution G>A against the reference sequence at position 5691 is identifiable by the consistency (>90%) throughout all the sequenced reads. Similar sequence variations were also observed for the other two reconstructed genomes, ToMMV and PepMV (see below).

### 2.4. ToBRFV Full-Length Genome Characterization and Phylogenetic Analysis

The full-length genome sequence of ToBRFV (isolate Tom-BA21, submitted in GenBank with the accession number OK624678), 6392-nt long, was assembled. The genome organization was identical to that shown by other isolates of the same virus.

Four ORFs were identified. The first, which starts at position 75 and ends at position 3425, and the second, which spans from positions 3501 to 4922, are predicted to translate into the two RdRP subunits, indicated as ORF1 (1116 aa) and ORF2 (473 aa), respectively. The third ORF, starting at position 4909 and ending at position 5709, was predicted to encode the 266 aa MP, whereas the fourth coding sequence (nucleotides 5712 to 6192) putatively translates into the CP (159 aa long) (Figure 3). The full-length sequence showed the highest identity (6381 matching nucleotides out of 6392, with a 99.82% identity) with the ToBRFV-Israeli isolate (ToBRFV-IL, acc. no. KX619418.1). At the aminoacidic level, ToBRFV-Tom-BA21 shared with the same ToBRFV-IL isolate an almost complete identity, differing by two, one, or zero amino acids from RdRP-ORF1, MP, and CP, respectively.

In order to trace back a possible origin of the newly identified ToBRFV isolate, a phylogenetic analysis using all the full-length non-redundant genomes available in GenBank database was run. Since the complete phylogenetic tree was not so informative due to the massive presence of highly similar sequences originating from the same locations, a subset of entries from 56 representative accessions was selected. The results of the analysis suggested that ToBRFV full-length sequences, although highly similar, can be divided into two main phylogenetic groups, plus a third group with a very limited number of entries (Figure 4), in agreement with a previous analysis of the ToBRFV population structure [28]. The Italian ToBRFV isolate Tom-BA21 under study clustered in the larger Group 2, close to the genomes of isolates 2020015323_A from the United Kingdom and ToBRFV-CH from Switzerland.

### 2.5. Nanopore Sequencing with ToBRFV Specific Primers Detects the ToMMV Genome

We tested whether the use of ToBRFV specific primers could be selective for that virus when used for nanopore sequencing, or alternatively could allow the sequencing reaction of other similar viruses. To this purpose, nucleic acid extracts from tomato plants infected with ToMMV were processed for MinION sequencing. The Mexican isolate ToMMV-MX5 (GenBank acc. no. NC_022230), that shares a 78.73% sequence similarity (5037/6398 nt) with ToBRFV-Tom-BA21 (Appendix A), was used as the reference genome. Some of the six reverse primers show complementarity to the ToMMV-MX5 genome. In particular, RP6 anneals to the ToBRFV 3′ terminus that is 100% identical to ToMMV, and RP3 3′-terminal 19 nt are also perfectly complementary to the ToMMV RNA. To some extent (14/15 nt identity at the 3′-half of the sequence) RP5 could also prime the reverse transcription reaction on the ToMMV template (Appendix A). Most probably as the consequence of those sequence similarities, nanopore sequencing yielded 7264 reads (almost 5% of total reads obtained from ToMMV-infected plants, Table 2) mapping to the ToMMV-MX5 reference genome (Figure 5A). The ToMMV-specific reads were assembled and a consensus sequence with a 761X of coverage was obtained, that displayed a 99.0% (6352 of 6398 nt) similarity with ToMMV-MX5 (Figure 5A). A BLAST research found an even higher 99.88% similarity with ToMMV-CpB1 from Brazil (GenBank acc. no. MH128145). These results indicate that, although the priming process may be less efficient, ONT sequencing with ToBRFV-specific primers will detect a different tobamovirus species, providing full coverage of its sequence and easy genome assembly.

### 2.6. PepMV Sequences Identification and Full-Length Genome Analysis

From the ONT sequencing of RNA extracted from ToBRFV-infected tomato plants, a set of 3281 (2.52% of total) reads was retrieved that showed high sequence similarity to PepMV and a coverage of 1606.9×. The latter virus had not been detected previously on the same plant samples before sequencing. Although the number of sequenced reads was much lower than those mapping to ToBRFV, for which specific primers were used, an almost complete reconstruction of the genome of the PepMV isolate coinfecting tomato plants was still possible.

The alignment of sequenced fragments to the PepMV-CAMP1-10 isolate RNA, used as the reference genome, is shown in Figure 5B. The 3′ region of the genome, comprised between nucleotide 6168 and the end of the viral RNA, was poorly represented by ONT sequencing. The corresponding RT-PCR product was subjected to Sanger sequencing for genomic sequence completion. The full sequence of the newly sequenced Italian isolate, named PepMV-Tom-BA21 (GenBank acc. no. OL362110), displayed 98% nucleotide sequence similarity with PepMV-CAMP1-10 (GenBank acc. no. HG976946), PepMV-SIC1-09 (HQ663891), and PepMV-SIC2-09 (HQ663892), all isolates from Southern Italy, but also with PepMV-Ch2 (DQ000985), a Chilean isolate that also defines one of the four main PepMV genotypes [29].

## 3. Discussion

In this work, we developed an innovative nanopore sequencing procedure that successfully allowed sequence enrichment for the specific detection of ToBRFV, the target viral pathogen. The specificity of the assay was increased by introducing multiple sequence-specific primers in the reverse transcription step. This strategy highly enriched the amount of ToBRFV specific reads, that reached up to 30% of the total number of high-quality reads from infected tomato samples. However, the same ONT sequencing protocol was also able to provide a significant amount of sequences (nearly 5%) from a different tobamovirus, ToMMV. For highest specificity, the amplicon-based sequencing technology, where a preliminary PCR enrichment of the target genome is performed, provides the nearly exclusive identification of target sequences. For instance, for the identification of another tobamovirus, cucumber green mottle mosaic virus (CGMMV), Mackie and coauthors proposed short amplicons as the target enrichment step combined with MinION sequencing. That strategy was shown to be highly specific to the target virus, excluding other closely related tobamovirus species, as well as the host genome [30]. Conversely, the lowest specificity is attained with standard library preparation procedures that use either oligo-dT or random primers for cDNA synthesis [27,30,31]. In agreement with this notion, the inclusion of oligo-dT primers by Chalupowicz and colleagues in the reverse transcription reaction allowed the detection by ONT sequencing of ToBRFV and other RNA viruses in host plants, although only few virus-specific reads, in the number of hundreds or even less, were identified in infected tissues [30]. By using the non-target-specific VN primers (VNP) supplied by ONT, and coupling data from both ONT and Illumina sequencing platforms, Chanda and coauthors almost completely reconstructed the ToBRFV genome. However, the individual contribution of either platform to the final genome assembly was not specified [25]. Thus, we can conclude that the target-specific priming method offers an intermediate level of specificity, providing enrichment of ToBRFV sequences while also giving information on other unrelated viral species co-infecting the same samples. The proposed strategy overcomes previously reported low sensitivity of nanopore sequencing in viral metagenomic approaches in human samples [32,33].

Since we used multiple target-specific primers for reverse transcription, we could have expected a higher percentage of target-specific reads. Instead, the ToBRFV and the ToMMV libraries produced over 50% and 70% of host reads, respectively. Additionally, the former library contained 2.5% PepMV-related reads. Most likely, the generation of off-target sequences deriving from the host or other organisms is a consequence either of the nonspecific annealing of primers, or a priming activity of self-complementary regions of RNAs in the reverse transcription step [27].

ONT sequencing technology provided long sequence reads, that in the case of ToBRFV averaged 800 bases, from which complete viral genomes were reconstructed by mapping to reference. Although the MinION platform produces reads close to full-length viral genomes, the de novo assembly step is still a major challenge of these technologies [34]. De novo assemblers, like Canu and Flye, have recently been used in our lab with no success in viral metagenomics studies starting from dsRNA (random-primed cDNA-PCR) and direct RNA with ONT MinION [27]. The de novo assembly performed with either Canu or Flye did not allow the complete reconstruction of the viral genomes, with Canu performing better than Flye. This result suggests that even a deeper sequencing of the viral fraction cannot completely overcome the limitations posed by the algorithms of these two programs, since both of them are optimized for longer genomes and reads [35,36]. In the case of Canu, the length of the ToBRFV-derived contig (5932 nt) obtained is smaller than the longest read (6047 nt) retrieved during the sequencing and aligning to the full-length genome using Minimap2. Likely, the Canu algorithm has been enhanced to improve repeats and aplotype separation [37]. The fact that viruses exist in their hosts as a quasi-species showing intraspecific sequence variability [38] could affect the final result of the assembly of a virus genome, for instance producing small contigs and discarding reads having multiple SNPs or positions not sufficiently covered by a proper number of consistent reads.

When analyzing the consensus sequence obtained by Minimap2 mapping of the reads against the viral reference genomes, we could observe that, despite the nanopore technology intrinsic error rate (10–15%), all the SNPs identified were consistent and significant. The detected SNPs were covered by more than 90% of reads aligned in the specific position under analysis. The major limitations posed by the error rate were overtaken reconstructing the viral genomes by the mapping to reference strategy. The high coverage, attained by the targeted enrichment via specific primers, made it possible to obtain reliable master consensus sequences.

The use of specific primers for cDNA synthesis favored the production of shorter reads, but, even in the case of ToBRFV, for which the primers were highly specific, no full-length reads have been retrieved. Secondary structures in the viral RNA genomes, e.g., the presence of a tRNA-like structure at the 3′ end in the case of tobamoviruses [39], probably hampered the optimal enzymatic activity on such folded regions for cDNA synthesis, thus reducing the efficiency of the process and the yield of longer sequences. Moreover, we observed an uneven coverage of the genome with a peak at position 5180, corresponding to the downstream region complementary to primer RP5, thus suggesting a possible different priming efficiency of the different reverse primers used for the cDNA synthesis.

In conclusion, the results presented here confirm that, with the proposed modifications, nanopore sequencing technology can be beneficially applied to the detection and genome analysis of viral pathogens. The targeted sequencing of a specific virus species did not hamper the simultaneous detection of off-target viral agents. The latter observation is of particular importance, since it allows the identification of previously unrecognized pathogens. The rapid detection of mixed infections is not to be underestimated, due to their effects on the epidemiology and the evolutive process of viruses, and the increased severity of the diseases that may be induced in crop plants [14,40].

## 4. Materials and Methods

### 4.1. Sample Description

Samples, composed of both leaf and fruit tissues, were collected in May 2021 from cherry-type tomato plants growing in a commercial greenhouse in the province of Bari, Southern Italy. A limited number of plants showed symptoms suggesting ToBRFV infection, e.g., growth reduction, leaf mosaic, leaf distortion, yellowing, and wilting. Fruits showed converging brown spots of irregular shape and rugose surface (Appendix A). Plant material was immediately used for RNA extraction and molecular (RT-PCR and RT-qPCR) detection of ToBRFV. Samples were also collected from tomato plants mechanically inoculated with a similar tobamovirus species, a Brazilian isolate of ToMMV [41,42], grown in containment facilities.

### 4.2. Nucleic Acid Extraction and ToBRFV Molecular Detection

Total RNA was extracted from each selected sample, using a guanidium thiocyanate extraction buffer followed by purification on the RNeasy Plant Mini kit (Qiagen, Valencia, CA, USA). One gram of plant tissue was homogenized in 3 mL of extraction buffer (4 M guanidine thiocyanate, 0.2 M sodium acetate, pH 5.0, 25 mM EDTA, 2.5% (*w*/*v*), and polyvinylpyrrolidone (PVP) and 400 μL of 20% N-Lauroylsarcosine sodium. One mL of each sample was incubated at 65 °C for 10 min and then immediately cooled on ice. After centrifugation at 12,000× *g* for 2 min, 500 μL of supernatant were loaded on the QIAshredder spin column and RNAs were sequentially extracted by RNeasy Plant Mini Kit (QIAGEN, Valencia, CA, USA) following the manufacturer’s instructions.

A first assessment for the ToBRFV occurrence in infected samples was performed according to the EPPO conventional RT-PCR-based protocol, using primers ToBRFV-F/ToBRFV-R developed by Alkowni and coauthors [13]. For further confirmation, one step duplex TaqMan real-time RT-PCR test was set up using CaTa28 and CSP1325 primers and probe as detailed in the EPPO standards [12]. The reaction was set up in 20 μL containing 10 μL of 2× TaqMan Fast Advanced Master Mix (Applied Biosystems) and cycling conditions were the following: reverse transcription at 50 °C for 10 min; denaturation at 95 °C for 3 min; 40 cycles of denaturation at 95 °C for 10 s and annealing and elongation at 60 °C for 60 s.

All the primers used in this work are listed in Appendix A.

### 4.3. Primer Design Strategy and Oxford Nanopore Technologies (ONT) Sequencing

RNA extracts previously employed in the RT-qPCR analysis were used for nanopore sequencing. Two libraries for sequencing were produced starting from RNA extracted from both the ToBRFV- and the ToMMV-infected tomato plants.

For increasing the possibility to fully cover the ToBRFV genome by sequencing, six reverse primers pairing at regular intervals of about 1000 bases were manually identified. The primers, designed on the isolate ToBRFV-IL genome sequence (GenBank acc. no. KX619418), were complementary to the following regions: RP1 (nucleotides 972 to 991); RP2 (1999–2019); RP3 (2965–2984); RP4 (3990–4009); RP5 (5182–5201); RP6 (6373–6392) (Appendix A; Appendix A; Figure 2).

About 100 ng total RNA per each one of the two libraries was quantified (Implen™ NanoPhotometer™ N60 Micro-Volume UV-VIS, München Germany), mixed with 1 μL 10 μM dNTP and 1 μL of the six reverse primers mixture at 10 µM each. The reaction was incubated at 98 °C for 3 min. First strand cDNA reaction was set up with Maxima H Minus Reverse Transcriptase (200 U/μL, Thermo Fisher Scientific), including 1 μM Strand-Switching Primer (SSP, supplied in the SQK-PCS109 cDNA PCR sequencing kit (ONT, UK)) according to manufacturer’s instructions. Thermal conditions for cDNA synthesis and strand switching were as published [31]. Reverse-transcribed sample was amplified by PCR with LongAmp Taq 2× Master Mix (New England Biolabs, Ipswich, MA, USA) and then purified twice with 0.8× volumes of resuspended AMPure XP beads (Beckman Coulter, Brea, CA, USA) following the manufacturer’s instructions. The final library was incubated at 37 °C for 15 min for increasing the yield, eluted in 14 μL of ONT Elution Buffer, and quantified and analyzed both for size and quality on agarose gel electrophoresis. Finally, library tied to rapid adapter (RAP) was loaded onto the MinION flow cell (Flo-Min 106d r9.4.1) for a 10 h run.

### 4.4. Sanger Sequencing of the Viral Genomes Terminal Regions

PCR amplicons obtained with different primer pairs (listed in Appendix A) were generated in order to establish the nucleotide sequences of the 5′- and/or 3′-terminal regions of the viruses under study. The amplicons were subsequently purified using a PCR Purification Kit (Qiagen, Valencia, CA, USA) and directly sequenced by using the same primers using the standard Sanger technology.

### 4.5. ONT Datasets Analysis

The general quality of the sequencing was assessed and evaluated using both the MinKNOW software included in the ONT-minit-release v. 19.06.9 (Oxford Nanopore Technologies, UK), and PycoQC [43]. Raw reads obtained from the ONT sequencing that passed the MinKNOW quality filtering were quality checked and demultiplexed using the “Fastq Barcoding” workflow embedded in the EPI2ME Desktop Agent Program (Oxford Nanopore Technologies, UK). Each set of demultiplexed reads were used for de novo assembly with Canu v2.1 [37] and with Flye v2.8.3 [36].

Obtained contigs were annotated using nucleotide database or viral sequence custom database via BLASTn and BLASTx algorithms, respectively. Blast results were considered significant when e-value thresholds were below 10–6 and 10–4 for BLASTn and BLASTx outputs, respectively.

Once the viral genomes in each library were identified, mapping to the complete viral genome sequences used as the reference was run using Minimap2 aligner program [44]. The final consensus sequences were extracted from the alignment after mapping the corresponding reads to the closest reference viral genomes using Geneious Prime v. 2021.2 (Biomatters, San Diego, CA, USA).

### 4.6. Phylogenetic Analysis

The evolutionary history was inferred by using the Maximum Likelihood method and Hasegawa-Kishino-Yano (HKY) model [45]. The tree with the highest log likelihood (−15,150.46) is shown. Initial tree(s) for the heuristic search were obtained automatically by applying Neighbor-Join and BioNJ algorithms to a matrix of pairwise distances estimated using the Maximum Composite Likelihood (MCL) approach. A discrete Gamma distribution was used to model evolutionary rate differences among sites [5 categories (GTR+G, parameter = 0.2490)]. Evolutionary analyses were conducted in MEGA X [46].

## Figures and Tables

**Figure 1 plants-12-00999-f001:**
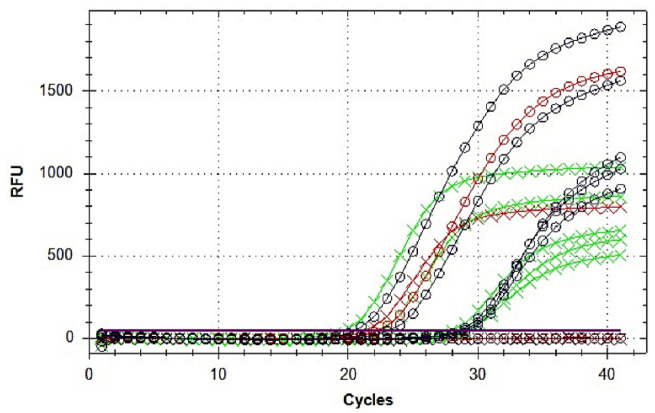
Duplex TaqMan real-time RT-PCR amplification plots for the detection of ToBRFV in five individual tomato plants. Black and green curves show amplification from symptomatic tomato samples. Red curves, positive (total nucleic acid extracted from infected tomato tissues) controls. Brown curves, negative (no template nucleic acid) controls. Black circle (O) and green cross (X) symbols indicate amplification curves with primer/probes CaTa28/FAM and CSP1325/ Cy5, respectively.

**Figure 2 plants-12-00999-f002:**
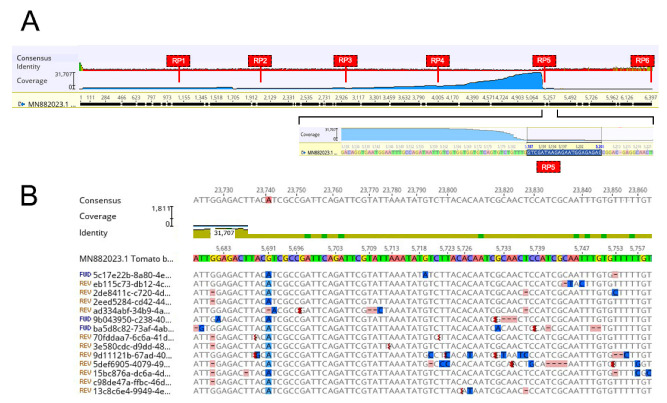
Coverage and nucleotide sequence variability within individual Oxford Nanopore Technology cDNA reads, from virus-infected tomato plants, using tomato brown rugose fruit virus (ToBRFV), Dutch isolate (MN882023.1) as the reference genome. (**A**) Coverage of MinION-derived reads within the full-length sequence of the ToBRFV genome, represented by the blue area. The position of the six reverse primers (RP1-6) on the ToBRFV genome is shown as red labels. In the insert, the detailed position of RP5 complementarity. Vertical bars on the left indicate the maximum coverage of nucleotide positions expressed in X. (**B**) Example of alignment of MinION-derived reads on the ToBRFV genome region 5678–5759. A number of low-frequency individual sequence variations are shown (blue shading). At position 5691 all reads show a consistent sequence variation that indicates a G>A substitution as compared to the reference sequence. Sequence and alignment visualization was performed by Geneious Prime.

**Figure 3 plants-12-00999-f003:**
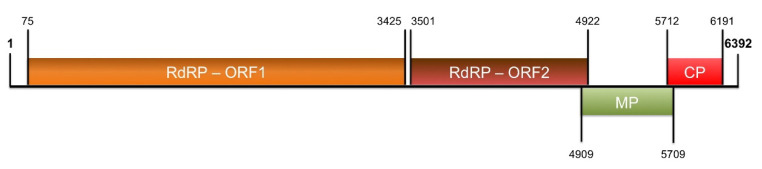
Sequencing analysis using ONT MinION. A graphical representation of the genome organization of ToBRFV isolate Tom-BA21. RdRp: RNA-dependent RNA polymerase; ORF: open reading frame; MP: movement protein; CP: coat protein.

**Figure 4 plants-12-00999-f004:**
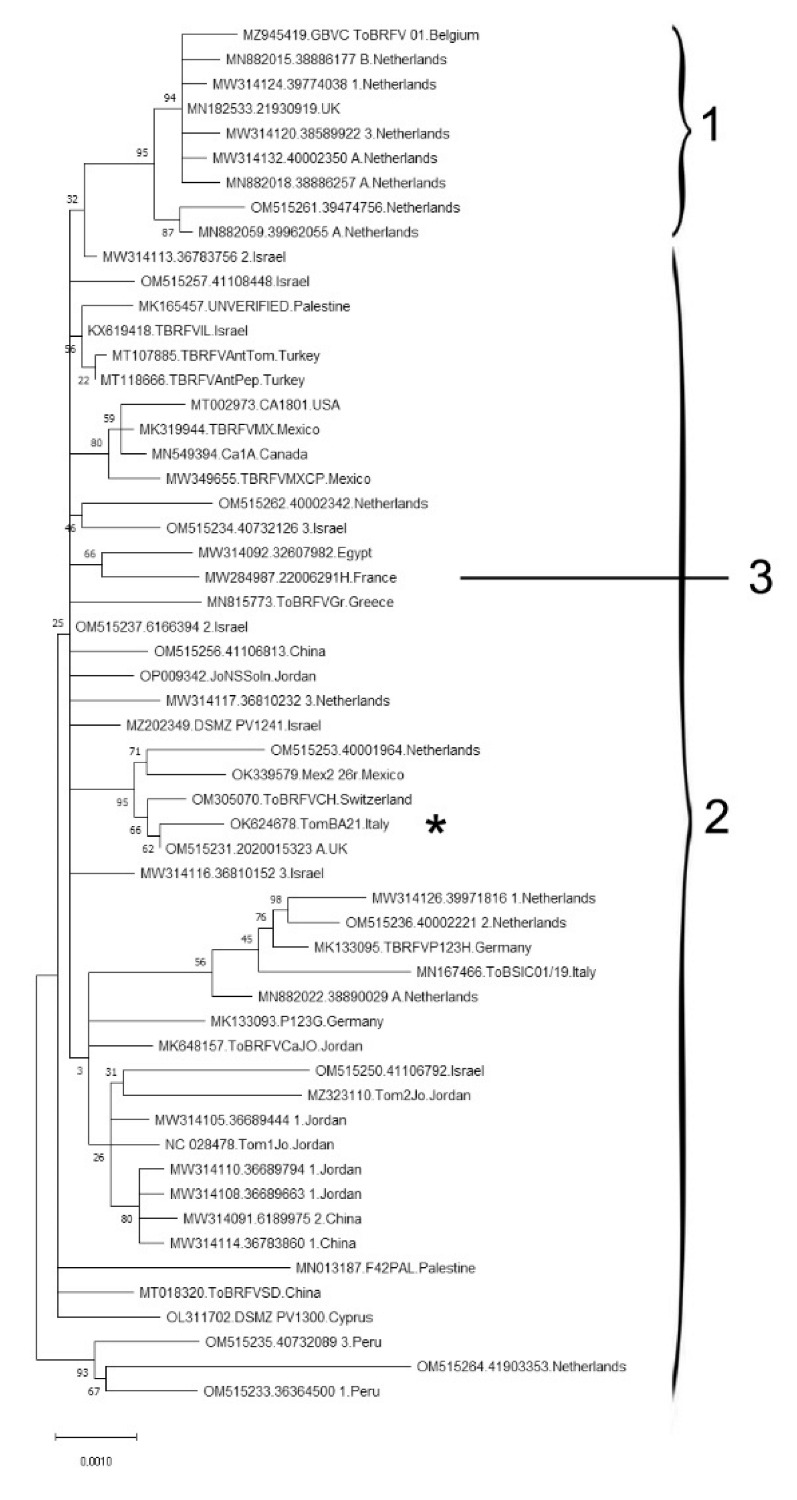
Maximum-likelihood phylogenetic tree by using the method with HKY+G model and 1000 replicates bootstrap value. The distribution of 56 ToBRFV isolates, based on their complete genome sequences, is shown. The asterisk marks the position in the tree of the Italian isolate ToBRFV-Tom-BA21 under study. The sequence of a different tobamovirus species, ToMMV-MX5 (GenBank acc. no. NC_022230), was used as the outgroup to root the tree MEGA X software (not shown in the figure). Numbering on the right side of the graph refers to the three ToBRFV phylogenetic groups as described by Çelik and coauthors [28].

**Figure 5 plants-12-00999-f005:**
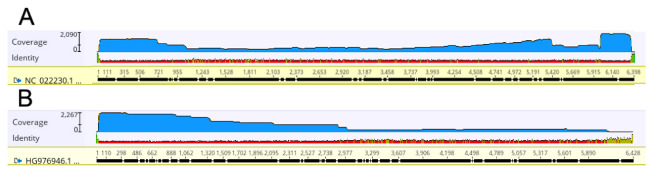
Coverage of MinION-derived reads aligned to the full-length sequences of (**A**) the ToMMV genome, and (**B**) the PepMV genome. The coverage of reads mapping along the two viral genomes is graphically represented by the blue areas. Vertical bars on the left indicate the maximum coverage of nucleotide positions expressed in X.

**Table 1 plants-12-00999-t001:** De novo assembly with the Flye and Canu tools.

	Flye		Canu	
	Blastn Annotation	Contig Length (bp)	Blastn Annotation	Contig Length (bp)
ToMMV (Bar03)	Unknown	19,102	ToMMV	5941
	Unknown	9070	*S. lycopersicum*	2765
ToBRFV (Bar04)	ToBRFV	615	ToBRFV	5932
	ToBRFV	699		
	*S. lycopersicum*	12,778		
	PepMV	8482		

**Table 2 plants-12-00999-t002:** Summary table of reads aligned to reference sequences.

Library	Ref. Sequence (Acc. No.)	Nr of Mapping Reads	% out of Total Quality Filtered Reads
ToBRFV (Bar04)	*S. lycopersicum* genome (assembly SL3.0)	74,432	57.24
	*S. lycopersicum* chloroplast (NC_007898.3)	145	0.11
	*S. lycopersicum* mitochondrion (NC_035963.1)	58	0.04
	ToBRFV (MN882023)	40,114	30.85
	PepMV (HG976946)	3281	2.52
	TOTAL	118,030	90.77
ToMMV (Bar03)	*S. lycopersicum* genome (assembly SL3.0)	114,042	76.83
	*S. lycopersicum* chloroplast (NC_007898.3)	2542	1.71
	*S. lycopersicum* mitochondrion (NC_035963.1)	68	0.04
	ToMMV (NC_022230)	7264	4.89
	TOTAL	123,916	83.48

## Data Availability

The full-length genomes of ToBRFV and PepMV isolates described in the present work have been deposited in GenBank, as indicated in the main text. All the sequencing output dataset generated in the study is freely available upon request to the Corresponding Author.

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
