# Peer review of "Nanopore Technology Applied to Targeted Detection of Tomato Brown Rugose Fruit Virus Allows Sequencing of Related Viruses and the Diagnosis of Mixed Infections"

_plants, 2023, doi:10.3390/plants12050999_

Round 1
Reviewer 1 Report
The manuscript by Kubaa et al. describes the use of nanopore technology for the detection and sequencing of ToBRFV in tomato plants. The disease produced by ToBRFV constitutes a serious threat to tomato production, so its early identification in tomato greenhouses is of special importance. The work presented has allow to detect and sequence in a fast way a sample with ToBRFV-like symptoms, identifying the virus causing the disease. This approach can be really interesting as a way of fast detection in field samples. However, there are some minor points that I would like to comment to the authors:
General comments
Please, try to improve the editing of English language, there are some sentences that are difficult to understand and misspelled.
In my opinion, one of the conclusions of this study is that ONT sequencing is not really specific and it is not a good option if you want to obtain a reliable sequence of one virus in mixed infections. As authors obtained contigs from ToMMV and PepMV with specific primers for ToBRFV, they can’t be sure in mixed infections that reads of one virus are not assembling with the reads of other virus, especially considering the low reliability of ONT sequencing. It is true that 800nt reads should be enough to differentiate between viruses, but between very similar viruses it may not be. So, although ONT can be a good option for detection, I think that sequences are not good enough to stablish conclusions in phylogenetic analyses. Authors just say that it is a good point to detect other viruses with ONT, but I think they should go deeper into this idea, or at least comment on it.
Authors should unify the reference list: there are references with page numbers and others without them, references with and without DOI, some capitalized words that shouldn’t be… In addition, please unify the references of the web pages; the web pages visited should include the date of visit.
Specific comments
Line 183. Authors should explain before in the text what are RPs and how many they use.
Lines 165-169. In Table 1 authors obtained a contig of 5,932 bp in the case of ToBRFV using Canu, but later they use an assembly of 6,392 bp (line 209). I suppose that this new assembly came from mapping against the reference accession, but it is not clearly explained. Please, add in the lines 165-169 the size of the new assembly obtained.
Figure 4. The image has poor quality, please try to improve it. In addition, authors should collapse the branches with less than 60 of support (at least).
Author Response
Reviewer #1:
General comments:
- Please, try to improve the editing of English language, there are some sentences that are difficult to understand and misspelled.
We improved the English language throughout the text, as indicated by both reviewers. We shortened sentences and opted for a direct and unambiguous language. Many sentences were eliminated for better readability of the whole manuscript.
- In my opinion, one of the conclusions of this study is that ONT sequencing is not really specific and it is not a good option if you want to obtain a reliable sequence of one virus in mixed infections.
In our study, we provided evidence of the reliable specificity obtained with the proposed nanopore sequencing approach. The use of ToBRFV-specific primers yielded a targeted virus enrichment up to 30% out of total reads. Either the ToBRFV or the off-target PepMV genomes were fully covered by overlapping reads with a coverage as high as 6800X and 761X, respectively. We also run a blastn/x annotation on individual reads, gaining robust confirmation of unequivocal identification of the two viruses. Additional information on this analytical process was added in paragraph 2.3
- As authors obtained contigs from ToMMV and PepMV with specific primers for ToBRFV, they can’t be sure in mixed infections that reads of one virus are not assembling with the reads of other virus, especially considering the low reliability of ONT sequencing.
The genomic sequences of ToMMV and PepMV, as well as ToBRFV, were obtained by the “mapping to reference” approach. The two heterologous viruses were first identified by blast search of individual contigs. Since the contigs were not covering the full-length genomes, a guided assembly through a mapping to reference was performed. Aligner programs (Minimap2 was used in this work) for long reads are based on probabilistic algorithms. To each read, a score based on the quality of the alignment to the reference sequence (different scoring for gaps, mismatches and alignment percentage) is given. If the score is below a certain threshold, the read is automatically excluded from the alignment. Therefore, misassignment of reads is highly unlikely, especially in the case of taxonomically unrelated viruses where sequence homology is null (e.g., PepMV vs. tobamoviruses). Hence, in our case study, although the error rate of the ONT sequencing technology is undeniable, this does not affect the virus identification at the species level, neither their mapping to reference since low quality, unrelated or partially related reads are automatically discarded.
- It is true that 800nt reads should be enough to differentiate between viruses, but between very similar viruses it may not be. So, although ONT can be a good option for detection, I think that sequences are not good enough to establish conclusions in phylogenetic analyses. Authors just say that it is a good point to detect other viruses with ONT, but I think they should go deeper into this idea, or at least comment on it.
The phylogenetic analysis presented in the paper is based on the consensus sequence obtained by mapping to reference, as explained above. Furthermore, as we discuss in the text, nucleotide changes detected in the consensus sequence are consistent among the mapping reads that provide high coverage of the viral sequences under analysis. The phylogenetic analysis was proposed as an indication of the possible origin of the new Italian isolate sequence (consensus sequence build and verified as above).
- Authors should unify the reference list: there are references with page numbers and others without them, references with and without DOI, some capitalized words that shouldn’t be… In addition, please unify the references of the web pages; the web pages visited should include the date of visit.
Thank you for this observation. The reference list was thoroughly revised in order to unify the format and complete all required fields.
Specific comments
- Line 183. Authors should explain before in the text what are RPs and how many they use.
We have added information about RPs in paragraph 2.2, page 5, as requested
- Lines 165-169. In Table 1 authors obtained a contig of 5,932 bp in the case of ToBRFV using Canu, but later they use an assembly of 6,392 bp (line 209). I suppose that this new assembly came from mapping against the reference accession, but it is not clearly explained. Please, add in the lines 165-169 the size of the new assembly obtained.
We corrected the text as correctly requested by the reviewer.
- Figure 4. The image has poor quality, please try to improve it. In addition, authors should collapse the branches with less than 60 of support (at least).
Figure 4 has been changed according to the reviewer’s suggestion, and its readability has been improved

Reviewer 2 Report
In this paper the authors study infection of tomato with tomato brown rugose fruit virus (ToBRFV), confirming its presence with RT-PCR and qPCR followed by a high-throughput sequencing using total RNA and nanopore ONT. Two libraries were synthesized by this innovative target enrichment technology, with 30% reads mapping to the virus. When the same set of primers applied to the ToMMV, a related virus, only 5% reads mapped to the latter virus. They also extracted a sequence of complete genome of nonrelated pepino mosaic virus (PepMV) suggesting that a low-rate off-target sequencing can inform about co-infecting viruses. Overall, the authors demonstrate that targeted ONT can both identify the targeted viral agents but also some non-targeted viruses in mixed infections.
This is an interesting piece of methodological research that should be published, but requires some deeper modifications, as follows.
1. Introduction part is too long and should be shortened roughly by half, retaining all the essential pieces of information.
2. Results:
-Sections 2.3. and 2.4. have similar titles, please better distinguish.
-Again, more compact English would improve the paper.
3. Discussion.
Discussion addresses all the important aspects of the nanopore sequencing strategy applied to the studied viruses. Although, again, I would shorten these writings, so the reader does not loose the sharp focus on the important cons and pros.
4. Methods, OK, but again some parts too detailed. Instead, appropriate references can be cited.
Author Response
Reviewer #2
- Introduction part is too long and should be shortened roughly by half, retaining all the essential pieces of information.
The Introduction part has been considerably shortened as requested, cutting out non-essential text.
- Results:
- -Sections 2.3. and 2.4. have similar titles, please better distinguish.
- -Again, more compact English would improve the paper.
Thank you for the first observation. Titles have now been changed as follows:
- 3 ToBRFV genome reconstruction by mapping of ONT reads
- 4 ToBRFV full-length genome characterization and phylogenetic analysis
The English language has been carefully revised, and sentences have been shortened for a more direct and clear communication.
- Discussion addresses all the important aspects of the nanopore sequencing strategy applied to the studied viruses. Although, again, I would shorten these writings, so the reader does not loose the sharp focus on the important cons and pros.
The discussion has been shortened, although some more explanation was added as requested by Reviewer 1
- Methods, OK, but again some parts too detailed. Instead, appropriate references can be cited.
Details have been eliminated in the text, and references added where appropriate.

Round 2
Reviewer 2 Report
The text has been abbreviated and now looks OK.